ecology, health and disease and epidemiology, computational biology

sea star wasting disease, mass mortality event, *Pycnopodia helianthoides*, temperature, species distribution models, echinoderm

**Author for correspondence:**
S. L. Hamilton
e-mail: hamiltsa@oregonstate.edu

# Disease-driven mass mortality event leads to widespread extirpation and variable recovery potential of a marine predator across the eastern Pacific

S. L. Hamilton[1], V. R. Saccomanno[2], W. N. Heady[2], A. L. Gehman[3,4], S. I. Lonhart[5], R. Beas-Luna[6], F. T. Francis[7], L. Lee[8,9], L. Rogers-Bennett[10,11], A. K. Salomon[12] and S. A. Gravem[1]

[1]Department of Integrative Biology, Oregon State University, Corvallis, OR 97331-4501, USA
[2]The Nature Conservancy, San Francisco, CA, USA
[3]University of British Columbia, Vancouver, BC V6T 1Z4, Canada
[4]The Hakai Institute, Campbell River, British Columbia, Canada
[5]NOAA's Monterey Bay National Marine Sanctuary, Monterey, CA, USA
[6]Universidad Autónoma de Baja California, Mexicali, Baja CA, Mexico
[7]Fisheries and Oceans Canada, Ottawa, Ontario, Canada
[8]Gwaii Haanas National Park Reserve, National Marine Conservation Area Reserve, and Haida Heritage Site, Parks Canada, British Columbia, Canada
[9]University of Victoria, Victoria, British Columbia, Canada
[10]Bodega Marine Laboratory, University of California Davis, Davis, CA, USA
[11]California Department of Fish and Wildlife, CA, USA
[12]Simon Fraser University, BC V5A 1S6, Canada

SLH, 0000-0002-0156-7610; RB, 0000-0002-7266-3394

The prevalence of disease-driven mass mortality events is increasing, but our understanding of spatial variation in their magnitude, timing and triggers are often poorly resolved. Here, we use a novel range-wide dataset comprised 48 810 surveys to quantify how sea star wasting disease affected *Pycnopodia helianthoides*, the sunflower sea star, across its range from Baja California, Mexico to the Aleutian Islands, USA. We found that the outbreak occurred more rapidly, killed a greater percentage of the population and left fewer survivors in the southern half of the species's range. *Pycnopodia* now appears to be functionally extinct (greater than 99.2% declines) from Baja California, Mexico to Cape Flattery, Washington, USA and exhibited severe declines (greater than 87.8%) from the Salish Sea to the Gulf of Alaska. The importance of temperature in predicting *Pycnopodia* distribution rose more than fourfold after the outbreak, suggesting latitudinal variation in outbreak severity may stem from an interaction between disease severity and warmer waters. We found no evidence of population recovery in the years since the outbreak. Natural recovery in the southern half of the range is unlikely over the short term. Thus, assisted recovery will probably be required to restore the functional role of this predator on ecologically relevant time scales.

## 1. Introduction

While the prevalence of mass mortality events (MMEs) is increasing with climate change [1,2], spatial variation in their timing, magnitude and triggers often remain unknown rendering recovery potential difficult to predict and conservation interventions challenging to design. MMEs constitute ecological disasters, and when they involve the loss of strongly interacting predators or foundation species, effects can propagate throughout ecosystems. In coastal

marine ecosystems, echinoderms, such as sea urchins and sea stars, appear particularly susceptible to disease-driven MMEs [3,4]. Furthermore, many echinoderm species are strong ecological interactors as predators or major grazers in their systems. Little is known, however, about the inter-actions between echinoderm disease and changing ocean conditions, making it difficult to determine when and where these collapses may occur (but see [5,6]). Our limited understanding of echinoderm disease-driven MMEs leaves us unprepared to respond to events that can rapidly alter population, community and ecosystem dynamics at continental scales.

The sea star wasting disease (SSWD) epidemic, also known as sea star wasting syndrome or asteroid idiopathic wasting syndrome, began in 2013 and affected over 20 species of sea stars along with the Pacific coastline from Mexico to the Aleu-tian Islands [7,8]. Previous outbreaks of putative SSWD have occurred, particularly in southern California, but have never impacted stars on the scale observed since 2013 [4]. *Pycnopodia helianthoides* (hereafter *Pycnopodia*) appears to be the species most impacted by SSWD, with declines reaching 99–100% in some areas [6,9–11]. Prior to the outbreak, *Pycnopodia* was recognized as an important generalist meso-predator across northeastern Pacific near-shore food webs and can be an effective predator of small- and medium-sized sea urchins on rocky reefs [12,13]. Via top-down pressure on sea urchins, *Pycnopodia* can promote kelp abundance by affect-ing sea urchin abundance, behaviour and grazing rates, although the strength of this phenomenon varies substantially across their range [10,12–14].

The aetiological agent(s) driving SSWD remain unidenti-fied. Current hypotheses focus on (i) a viral-sized aetiological agent (e.g. sea star-associated densovirus) and (ii) low oxygen at the surface of the skin maintained through subsequent bac-terial proliferation [7,15]. Additionally, the relationship between temperature and SSWD is unresolved. In laboratory studies, the lesion growth rate increased with increasing tem-perature, but evidence for warm temperatures triggering SSWD is mixed [16–18]. Some studies showed a positive relationship between the timing of the outbreak and temperature [6,18,19], while others found no relationship [8,20] or a negative relation-ship [21]. Differences in disease detection could explain these variable field observations. SSWD is a fast-paced disease accel-erating at the scale of weeks to months, so peak prevalence of infection is difficult to detect from seasonal or annual monitoring programmes [7]. Thus, the relationship between environmental triggers of an outbreak can easily be confounded with pandemic disease dynamics [22].

While previous papers have documented that SSWD caused dramatic losses in *Pycnopodia* in some places [7,9,10], here we compiled 48 810 surveys on *Pycnopodia* pres-ence and density from 34 data contributors ranging from Baja California, Mexico, to the Aleutian Islands, USA, to create the most comprehensive dataset to date to quantify impacts to the species across its entire range. Using this unique data-set, we evaluate the population-level impacts of SSWD on *Pycnopodia* by asking the following. (i) How did the timing of the SSWD epidemic vary across *Pycnopodia*'s range? (ii) How did SSWD change the abundance and spatial distri-bution of *Pycnopodia*? (iii) How did environmental variables that predict *Pycnopodia* distribution differ pre- and post-out-break? (iv) Is there evidence of population recovery in the years since populations first collapsed?

## 2. Methods

### (a) Data collection and compilation

Thirty research groups from Canada, the United States, Mexico, including First Nations, shared 34 datasets containing field sur-veys of *Pycnopodia* (electronic supplementary material, table S1). The data included 48 810 surveys from 1967 to 2020 derived from trawls, remotely operated vehicles, scuba dives and interti-dal surveys. We compiled survey data into a standardized format that included at minimum the coordinates, date, depth, area sur-veyed and occurrence of *Pycnopodia* for each survey. When datasets contained more than one survey at a site in the same day (e.g. multiple transects), we divided the total *Pycnopodia* count in all surveys by the total survey area and averaged the latitude, longitude and depth as necessary. Using breaks in data coverage, political boundaries and biogeographic breaks, we assigned each survey to one of twelve regions: Aleutian Islands, west Gulf of Alaska (GOA), east GOA, southeast Alaska, British Columbia (excluding the Salish Sea), Salish Sea (including the Puget Sound), Washington outer coast (excluding the Puget Sound), Oregon, northern California, central Califor-nia, southern California and the Pacific coast of Baja California (electronic supplementary material, figure S1).

### (b) Timeline of epidemic and population declines

We developed two timelines to define (i) epidemic phases describ-ing how the epidemic progressed and (ii) population phases describing how *Pycnopodia* populations changed over time (electronic supplementary material, table S2).

#### (i) Epidemic phases

For each region, epidemic timelines were divided into four phases punctuated by three dates as follows: (i) pre-epidemic phase; (ii) date SSWD first observed; (iii) emerging epidemic phase; (iv) outbreak date; (v) epidemic phase; (vi) crash date and (vii) post-epidemic phase (electronic supplementary material, figure S2). To describe SSWD emergence, we used datasets from MARINe (electronic supplementary material, table S1) and queried the date of the first symptomatic sea star observed at 594 sites distributed from Baja California, Mexico, to the western GOA, USA (see http://data-products/sea-star-wasting/). We used observations for both *Pisaster ochraceus* and *Pycnopodia* because *P. ochraceus* has more observations than *Pycnopodia* enabling more accurate estimates of outbreak timing among regions ($n = 450$ and $n = 247$ sites, respectively). *P. ochraceus* showed a slightly earlier date of first observation than *Pycnopodia*, but the timelines were otherwise very similar (See electronic supplementary material, figure S3).

We defined 'date SSWD first observed' as the earliest record of a symptomatic *Pycnopodia* or *P. ochraceus* in each region (elec-tronic supplementary material, figure S2). This date defined the break between 'pre-epidemic' and 'emerging epidemic' phases. We defined 'outbreak date' by fitting a normal curve to the dis-tributions of dates when SSWD was first observed at each site and calculated the 10th percentile; this served as the break between 'emerging epidemic' and 'epidemic' phases. The 10th percentile was chosen because we reasoned that when 10% of sites show signs of SSWD, the disease has probably transitioned to an outbreak, rather than persisting as isolated cases of infec-tion. Further, our detection of disease at 10% of sites probably means the actual number of sites infected is much higher. The time elapsed between the 'date SSWD first observed' and the 'outbreak date' was considered the 'emerging epidemic' phase. As the epidemic progressed and *Pycnopodia* populations declined, we used trends in *Pycnopodia* occurrence (site-level presence or absence) to estimate 'crash date', defined as the date when the occurrence rate of *Pycnopodia* in a region decreased

by 75% from pre-outbreak levels. A 75% decline in occurrence was chosen because it is a substantial decline and because this threshold gave date estimates in all regions that were the most similar to the crash timelines reported elsewhere [8–10,12,21,23]. 'Crash date' defined the break between the 'epidemic' and the 'post-epidemic' phases.

We defined 'emergence duration' as the time elapsed between 'date SSWD first observed' and the 'outbreak date', which indicated how quickly the disease progressed in each region. The difference in time between the outbreak date and crash date in a region defined the 'epidemic duration'. For further details, see electronic supplementary material, figure S2.

### (ii) Population phases

To define the effect of SSWD on *Pycnopodia* populations, we delineated three population phases: historical, decline and current (electronic supplementary material, figure S2). The 'outbreak date' in each region (defined above) determined the break between the 'historical' and 'decline' phases. The 'current' period includes data from 2017 to 2020. Region-specific dates associated with the 'post-epidemic' phase were not used to define 'current' population phase because (i) not all regions are necessarily in the 'post-epidemic' phase (see electronic supplementary materials) and (ii) many regions had recent crash dates (e.g. 2018 for Alaskan regions) with limited data in the 'post-epidemic' phase. Population phases were used in density and occurrence analyses, species distribution models and remnant population analysis.

## (c) Influence of sea star wasting disease on global sunflower sea star populations

To determine how *Pycnopodia* has been affected by SSWD, we examined how density and occurrence varied with population phase and region. We compared historical and current populations (defined above) in each region when possible. We modelled deep (greater than 25 m depth) and shallow (less than or equal to 25 m depth) populations separately because *Pycnopodia* were much more common at depths less than or equal to 25 m, and data from deep depths were unavailable for most regions. We performed all models in R v. 4.0.0 and RStudio v. 1.2.5042 [24]. For density models, we built zero-inflated generalized linear models [25] of *Pycnopodia* counts, using $\log_{10}$ (area searched) as the offset variable, Poisson likelihoods and log link functions, fit by Type II sums of squares. For occurrence models, we constructed a generalized linear model [26] of *Pycnopodia* occurrence rate, using area searched as a covariate, binomial likelihoods and logit link functions, fit by Type II sums of squares. In some regions, low sample sizes led to low confidence in our estimates of occurrence and density, therefore we used grey shading in our tables to delineate values with low confidence. For further details on this modelling process and regional data limitations, see electronic supplementary materials.

## (d) Abiotic correlates of the population decline

We used MaxEnt species distribution models to (i) quantify abiotic conditions associated with *Pycnopodia* before and after SSWD and (ii) predict the distribution of remaining populations [27]. We created two MaxEnt models, one estimating the distribution of *Pycnopodia* prior to the SSWD outbreak (2009–2012) using 6206 observations and the other estimating the distribution of current populations (2017–2020) using 1702 observations. We used prior studies to select important abiotic variables [28,29] and eliminated highly correlated variables [30]. Abiotic variables in each model were the 90th percentile of sea surface temperature and mean chlorophyll from 2009 to 2012 and 2017 to 2020 for pre-outbreak and current models, respectively (NASA MODIS Aqua: https://oceancolor.gsfc.nasa.gov/data/aqua/), mean salinity from a

long-term climatology (NOAA: https://www.nodc.noaa.gov/OC5/regional_climate/), depth (NOAA ETOPO1: https://www.ngdc.noaa.gov/mgg/global/), and substrate type (UC Boulder dbSEABED: https://instaar.colorado.edu/~jenkinsc/dbseabed) (see electronic supplementary materials for further details).

Datasets were clipped to the study area, defined as 0–456 m depth (our deepest observation of *Pycnopodia*) from 112.637° W, 24.874° N (our southernmost observation) and 170.196° W, 52.508° N (our northernmost/westernmost observation) [31]. Google Earth Engine was used to create temperature and chlorophyll metrics from MODIS data, and all other analyses were completed in R Studio [24,32]. We used our compiled *Pycnopodia* dataset to create 5000 background points for each model that mirrored the spatial sampling bias of the data itself [30]. Using the package 'ENMeval', we chose to use linear and quadratic features and a regularization parameter = 1 based on combined information from the training and evaluation Area Under the Curve metrics and Akaike's information criterion (see electronic supplementary materials for further details) [33]. We adjusted the default average species probability parameter by calculating the average occurrence rate from the pre-outbreak (0.61%) and current periods (0.14%) from the compiled dataset [30].

## (e) Current status and recovery potential
### (i) Population density

To visualize changes in *Pycnopodia* density in shallow depths (less than 25 m) from historical (1987–outbreak date) to current populations (2017–2020), we used ArcGIS Pro 2.7 to generate a grid of 16 km$^2$ hexagonal cells across *Pycnopodia's* range. For each time period, we used a spatial join to nest the available density surveys within each cell (historical, $n = 3984$; current, $n = 1344$) and calculated mean density within each cell for both time periods. Jenks natural break classification was selected to symbolize density due to the high variance within the dataset.

### (ii) Remnant populations

To determine where persistent remnant *Pycnopodia* populations have been found since 2017, we used ArcGIS Pro 2.7 to generate a grid of 16 km$^2$ hexagonal cells along with *Pycnopodia's* range. We used a spatial join to nest the 6284 available surveys from shallow depths for 2017–2020 within each cell. We retained only those cells with surveys performed in at least three of the 4 years from 2017 to 2020. From these better-surveyed cells, we calculated the percentage of surveys with *Pycnopodia* occurrence, which indicated the persistence of the remnant population. Each cell was then classified as 'absent' = 0%, 'rare' = less than 25%, 'common' = less than 90% and 'very common' ≥90%. Note that this method does not evaluate remnant *Pycnopodia* population dynamics. Remnant populations designated as common or even very common using this method can include populations that are (i) unaffected by SSWD and stable, (ii) affected by SSWD yet stable or (iii) affected by SSWD and declining.

## 3. Results

## (a) Latitudinal gradients in epidemic timing

Epidemic timelines showed that the date of first SSWD observed occurred in 2013 for nearly all regions (figure 1*b*; electronic supplementary material, table S2). Emergence duration (orange bar in figure 1*b*) was notably variable among regions. In British Columbia, the Washington outer coast, all California regions and Baja California, SSWD became an 'outbreak' (approx. 10% sites infected) within a few weeks to two months. The emergence duration was nearly a year

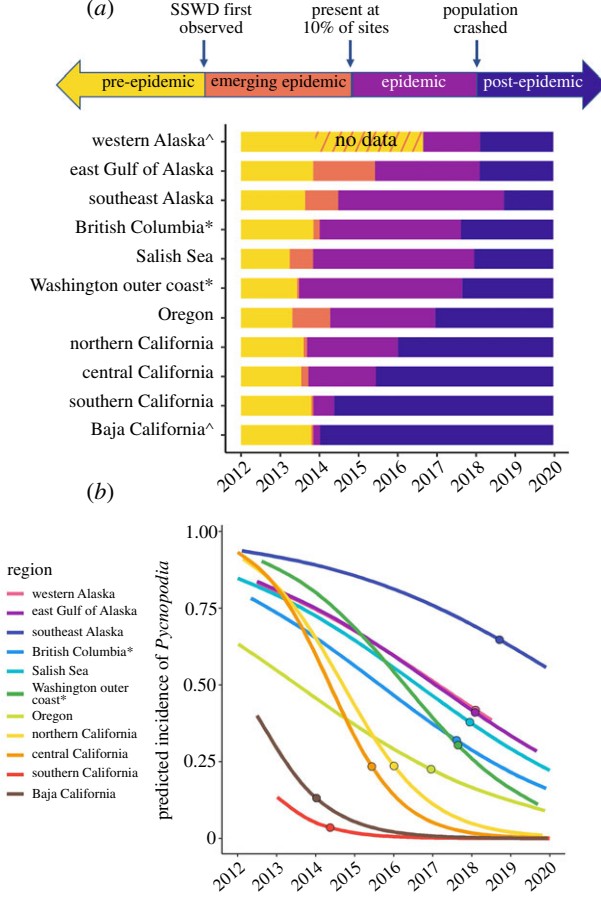

**Figure 1.** (*a*) Timeline of epidemic phases between January 2012 and December 2019 by region. Pre-epidemic phase (yellow) includes dates before the 'date SSWD first observed', when the first recorded symptomatic sea star was reported in each region (unknown in western Alaska). The emerging epidemic phase (orange) spans from the 'date SSWD first observed' to the 'outbreak date' when 10% of the sites within a region had reported SSWD observations. Epidemic phase (violet) spans the 'outbreak date' to the 'crash date' (defined above) and indicates how quickly the disease caused population declines. The post-epidemic phase (purple) includes dates after the crash date, though SSWD may still be present and driving further declines in the future. Caret: some dates inferred based on the dates in neighbouring regions. Asterisk: British Columbia and Washington outer coast exclude the Salish Sea. (*b*) Logistic model predictions for the occurrence of *Pycnopodia helianthoides* over the course of the epidemic by region. These models were used to estimate the 'crash date' (filled circles) of the populations in each region, defined as a 75% decline in occurrence from January 2012 to December 2019. (Online version in colour.)

in Oregon and over seven months in the Salish Sea, despite the Salish Sea having the earliest record of a SSWD-afflicted animal (30 March 2013). Southeast Alaska's emergence duration was similar to Oregon (10.1 months) but the emergence duration in the eastern GOA was nearly 19 months.

Epidemic duration (light purple bar in figure 1*a*) and the crash date (solid points in figure 1*b*) showed a marked latitudinal gradient, indicating that populations crashed more quickly in the southern part of the range (figure 1; electronic supplementary material, table S2). The logistic regression model showed significant declines in occurrence over time, which varied by region (electronic supplementary material, table S3). Populations crashed in Baja California within 2.1

months of the outbreak date and in southern California within 6.3 months. Declines took less than two years in central California, less than three years in northern California, Oregon and the east GOA, and around 4 years on the Washington outer coast, the Salish Sea, British Columbia and southeast Alaska. The west GOA and Aleutian Islands had an estimated 17-month epidemic duration, but limited sampling in these regions made these estimates uncertain.

For this analysis and others, lower data availability for much of Alaska and parts of British Columbia created greater uncertainty in regional estimates for these areas. We suspect, however, that the observed latitudinal gradient here is not driven only by generally lower sampling effort northward because northern regions with high sampling effort, such as southeast Alaska, also exhibited late outbreak dates and long emergence durations.

## (b) Latitudinal gradients in population declines

After the SSWD outbreak, *Pycnopodia* density declined range wide by 94.3% and the magnitude of this decline was similar in shallow and deep depths (92.5% and 96.5%, respectively, figure 2; electronic supplementary material, table S4). In shallow depths (where the vast majority of animals are found), the magnitude and significance of the decline differed by region (figure 2; electronic supplementary material, table S4 and table S5: population phase: $p = 0.423_{7,3523}$; region × population phase: $p < 0.0001_{7,3523}$). Estimated density declines were greater than 87.9% in 11 of 12 regions and were greater than 99.2% in all regions of the outer coast of the contiguous USA and Mexico, with no *Pycnopodia* observed in Oregon, southern California, and Baja California since at least 2017 (figure 3; electronic supplementary material, table S4). In the Salish Sea, the British Columbia, southeast Alaska and the east GOA, declines were also severe (92.4%, 87.9%, 96.0% and 93.8%, respectively).

Occurrence declined range wide by 52.3% (figure 2; electronic supplementary material, table S4), and this decline was significant in shallow and deep depths (64.13% and 55.3%, respectively; electronic supplementary material, table S5: $p_{1,3714} < 0.0001$ and $p_{1,2148} < 0.0001$, respectively). In shallow depths, regional patterns were similar to those for density declines (figures 2 and 3*a*,*b*; electronic supplementary material, table S4 and table S5: region × population phase: $p < 0.0001_{7,3714}$) with more severe declines in Oregon and southward (greater than 92.2% decline). In the Salish Sea, British Columbia, southeast Alaska and the east GOA, declines were substantial though less severe than southern regions (52.9%, 68.9%, 20.8% and 58.9%, respectively). Too few data were available to make confident estimates in the west GOA and the Aleutian Islands. Overall, *Pycnopodia* appears functionally extirpated along the southern 2700 km stretch of coastline from Baja California, Mexico, to Cape Flattery, Washington, USA, and experienced substantial declines in northern regions.

## (c) Temperature became more important in predicting *Pycnopodia* distributions

Prior to the outbreak of SSWD, MaxEnt models predicted a relatively even distribution of *Pycnopodia* from Baja California to the Aleutian Islands, and the predicted probability of *Pycnopodia* occurrence rarely dropped below 60% in coastal areas (figure 3*c*). Depth was by far the strongest predictor

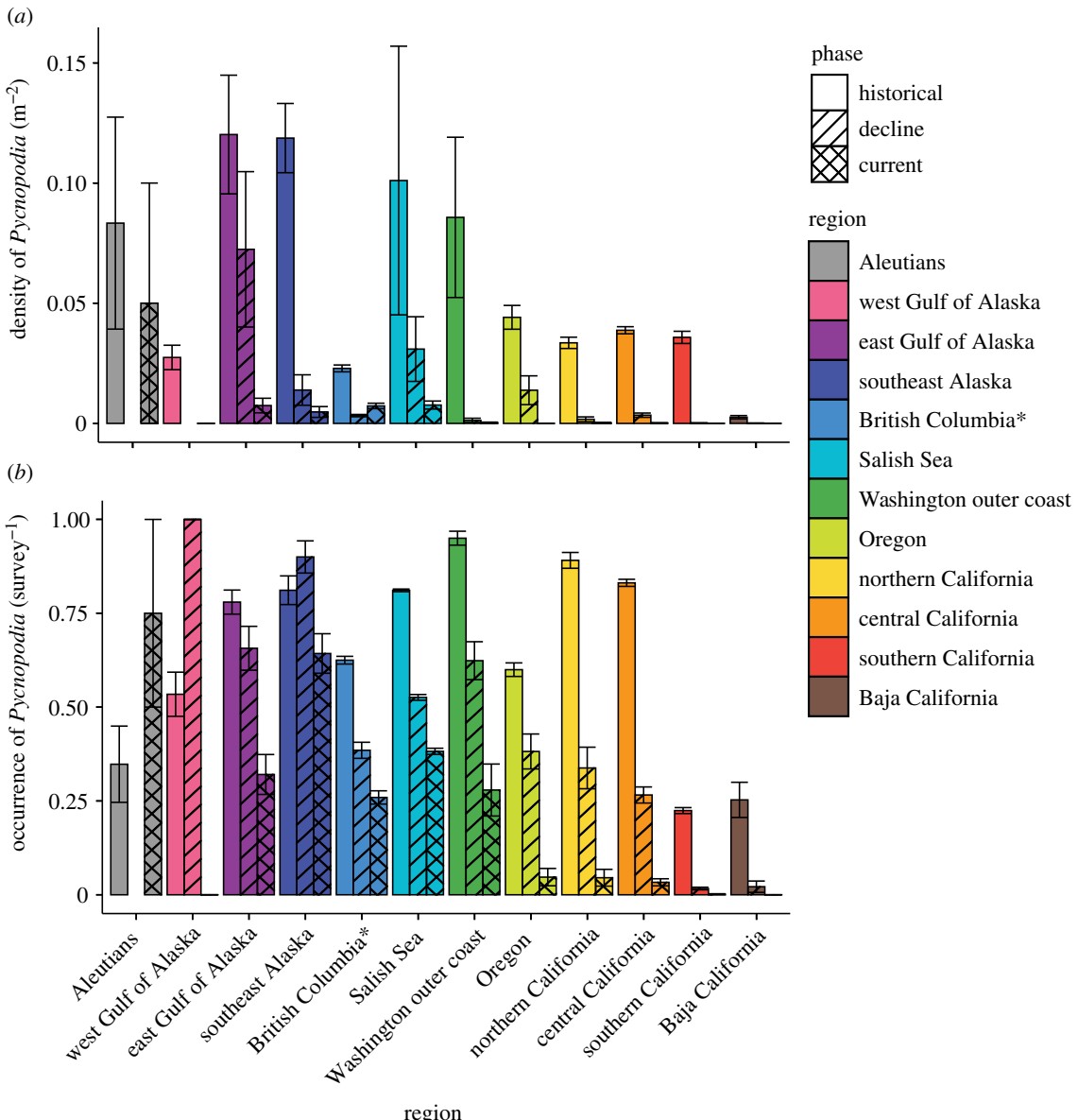

**Figure 2.** Mean (±s.e.) *Pycnopodia helianthoides* (*a*) density (m²) and (*b*) occurrence in shallow depths (less than 25 m) among the 12 regions and population decline phases (historical, decline and current, see electronic supplementary material, table S2) over the SSWD outbreak. Asterisk: Washington outer coast and British Columbia exclude the Salish Sea. (Online version in colour.)

of *Pycnopodia* occurrence with permutation importance of nearly 75% of the total predictive capacity (figure 4*a*; electronic supplementary material, table S6) [28]. The predicted probability of *Pycnopodia* dropped exponentially as depth increased, approaching zero around 300 m (figure 4*b*).

Compared to the pre-outbreak model, the probability of *Pycnopodia* occurrence plummeted range wide in the current model. MaxEnt models predicted nearly 0% probability in Baja California and southern California, and less than 10% probability across the outer coast of the US as far north as 48.4° latitude, around Cape Flattery, Washington (figure 3*d*). Moving northwards along inner coastal waters from Puget Sound to the Aleutian Islands, the current model predicted somewhat higher probabilities of occurrence around 15–25%. Along central British Columbia, southeast Alaska and the Aleutian Islands, the current model identified pockets of higher probabilities around 30–60% (figure 3*d*).

The importance of various abiotic variables in predicting *Pycnopodia* occurrence also differed between the pre-outbreak and current models. The importance of temperature increased

more than fourfold to nearly 40% permutation importance and was the most important predictor along with depth (figure 4*a*). Prior to the outbreak, the relationship between the probability of *Pycnopodia* and temperature formed a unimodal curve that peaked around 16°C (figure 4*b*). After the outbreak, this curve shifted dramatically towards colder temperatures, peaking around 5°C and decaying down to nearly 0% probability by 23°C (figure 4*b*). Conversely, depth maintained a similar relationship with predicted probability, although the peak at shallow depths fell to approximately 18% probability as opposed to approximately 75% pre-outbreak. Among the remaining variables, mean chlorophyll increased in importance to 10.7% permutation importance, substrate rose to 6.3% and mean salinity fell to become the least important variable (electronic supplementary material, table S6).

## (d) No population recovery since 2017

We found no clear evidence that *Pycnopodia* have begun to recover on a large scale. Though some sites have seen the

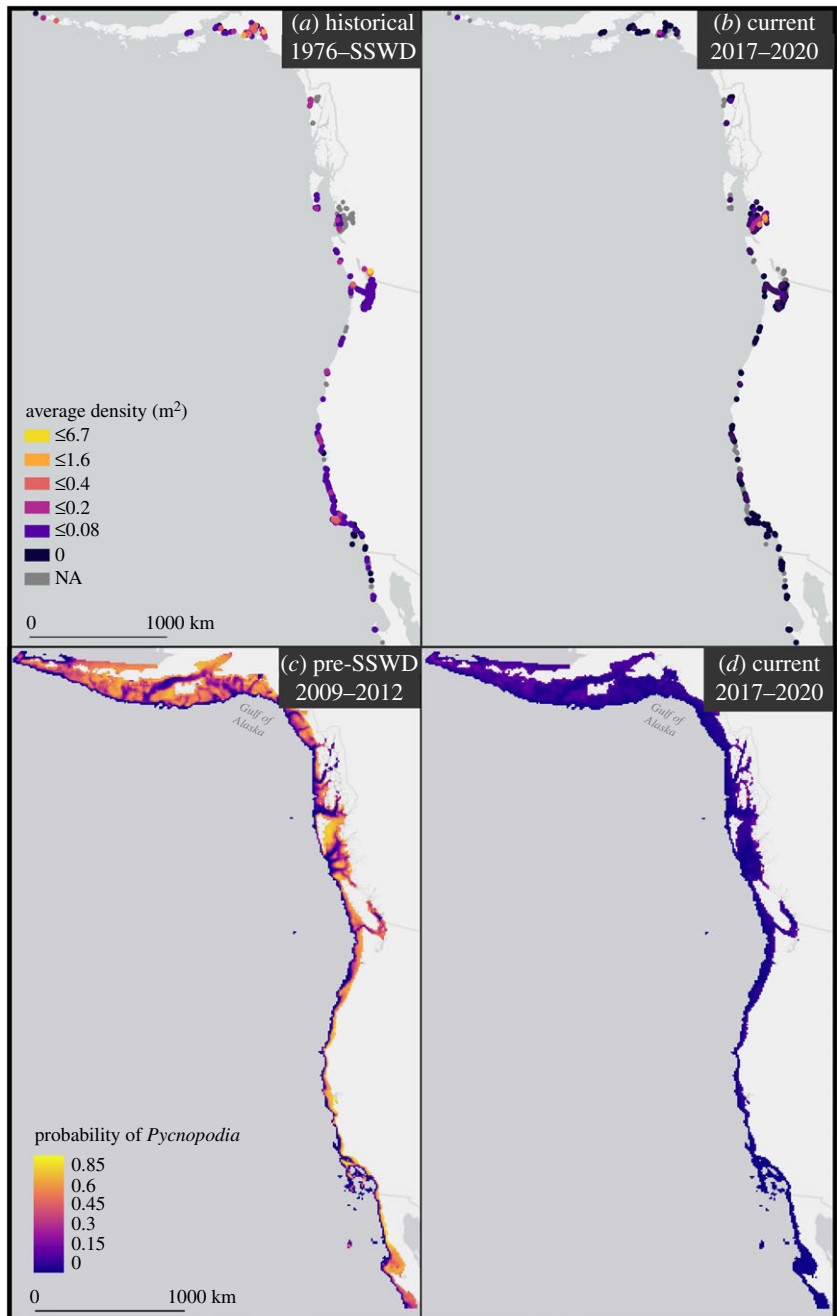

**Figure 3.** Density (m²) of *Pycnopodia helianthoides* in shallow water (less than 25 m) from (*a*) historical (1976 to the outbreak date of SSWS) and (*b*) current (2017–2020) surveys. Grey cells represent areas where no surveys were conducted during the relevant timeframe, but were conducted within the dataset timeframe. MaxEnt species distribution model logistic predictions for *Pycnopodia helianthoides* (*c*) immediately pre-SSWD outbreak (2009–2012) and (*d*) currently (2017–2020). (Online version in colour.)

recruitment of small animals (A.L.G. & S.A.G. 2017–2020, personal observation), we observed no increases in *Pycnopodia* density in any region since 2017 (electronic supplementary material, figure S4). In fact, the southern regions from Baja California to the outer coast of Washington have 'flat-lined' at near-zero densities. Further, those regions with remaining animals either show no recovery (east GOA) or a continued decline in density from 2017 to 2020 (southeast Alaska, British Columbia, Salish Sea; $p < 0.001$ for each region). However, fits by region were quite low ($R < 0.09$ in all regions) because the remaining densities in these regions were variable.

When we investigated localized (16 km²) persistence of remnant populations from 2017 to 2020, we found no cells with common or very common observations of *Pycnopodia* from Oregon to the southern range limit, and only two cells had

common populations on the Washington outer coast (figure 5). In the Salish Sea and north, the number of cells with common or very common observations increased, peaking at 60% of the cells in southeast Alaska. While the Aleutian Islands and west GOA had no regularly surveyed cells, we expect that common observations could be found there based on the increased probability of *Pycnopodia* in these regions predicted by the SDM models (figure 3) and cells with common observations in nearby regions of east GOA and southeast Alaska (figure 5).

## 4. Discussion

We document the functional extirpation of *Pycnopodia* across 2700 km of coastline from Baja California, Mexico to Cape Flattery, WA, USA and severe declines across the rest of their

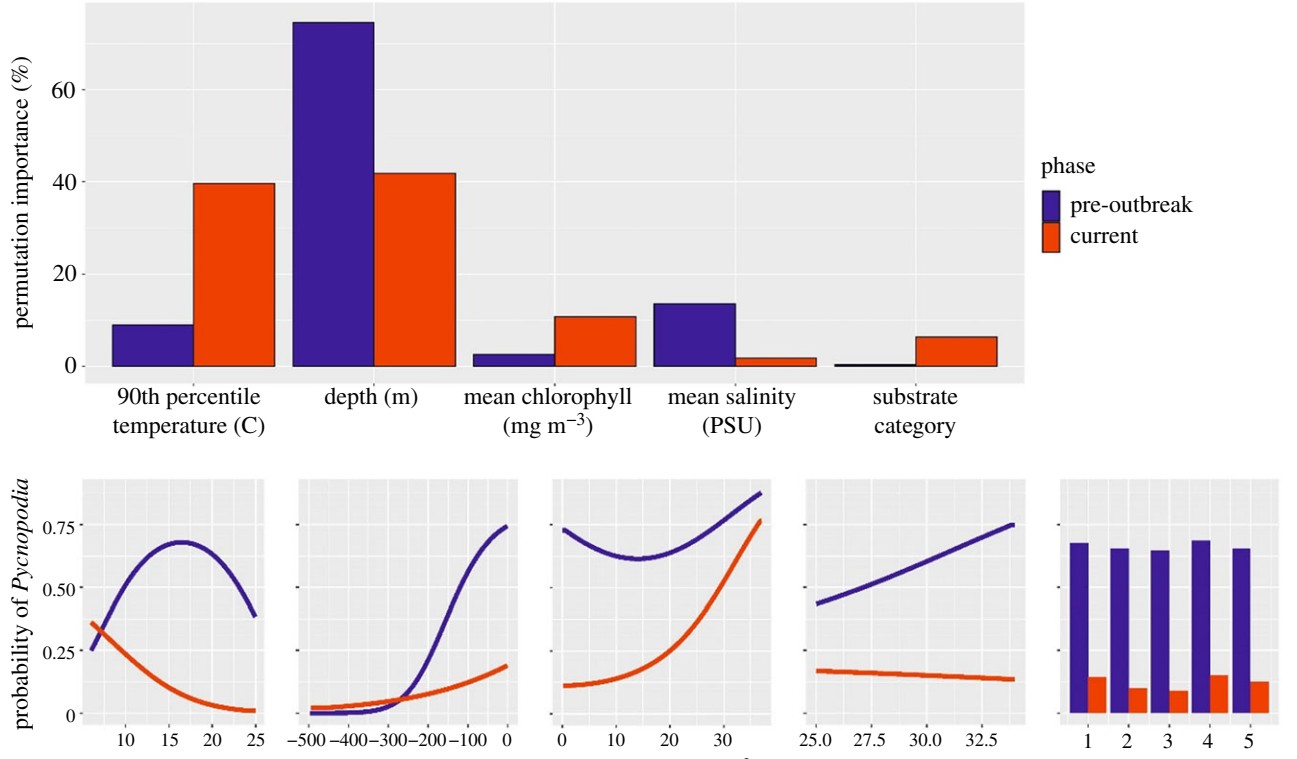

**Figure 4.** (*a*) Permutation importance of variables in MaxEnt model predictions *of Pycnopodia helianthoides* occurrence pre-outbreak (2009–2012) and current (2017–2020). (*b*) MaxEnt logistic output response curves showing the probability of *Pycnopodia* occurrence across the represented range of each variable pre-outbreak (2009–2012) and currently (2017–2020). (Online version in colour.)

range. Regions with warmer temperatures had faster, more severe population declines and fewer survivors. Currently, *Pycnopodia* populations show few signs of recovery, and populations in the northern half of the range may still be declining. The power of this analysis derived from the continental-scale collaboration that combined data from more than 30 contributors working across countries and sectors. If disease- and climate-driven MMEs continue to increase in frequency, this kind of multinational collaboration and data sharing will be critical to responding to these events, particularly for wide-ranging species like *Pycnopodia*. Our analysis sounds an urgent alarm for managers, policy-makers, conservationists and ocean-lovers across the Pacific Coast of North America. Without intervention, *Pycnopodia* are unlikely to recover to pre-wasting levels from Baja California to the outer coast of Washington in the near future. The persistence of the remnant populations throughout the rest of the range is also in question. Further, the widespread and potentially long-lasting loss of *Pycnopodia* may have ecosystem-level consequences, particularly for kelp forests, where this loss may erode their resilience via increased urchin grazing [10,12,13,34].

## (a) Latitudinal gradient in the speed and severity of sea star wasting disease

A strong latitudinal gradient structured the rate of regional *Pycnopodia* population crashes, suggesting that regional factors could be driving variation in disease response. Populations crashed within a few months in Baja California and southern California, 2 years in the rest of California and in 3–5 years in Oregon and northward. Populations may still be experiencing declines throughout Alaska (figure 1*b*), which is supported by

ongoing evidence of diseased *Pycnopodia* in many regions (P. Raimondi & K. Gavenus 2021, personal communication). The increased rate of disease spread in the southern latitudes suggests that environmental conditions either increased host susceptibility and/or disease transmission, or that genetic variability in the host or disease leads to a higher transmission rate (e.g. [35]). It will be difficult to disentangle these possibilities until a causative agent of SSWD has been identified.

The severity of SSWD-driven population declines also showed a marked latitudinal pattern. *Pycnopodia* populations appear to be approaching functional extirpation from Baja California, Mexico, to Cape Flattery, WA, USA. In our dataset, no *Pycnopodia* were observed in Baja California since 2015, none in California since 2018, and only a handful in Oregon and the Washington outer coast since 2018 (for more detail see [11]). In the Salish Sea and northward, *Pycnopodia* populations experienced severe declines but the chance of encountering an individual during a survey is greater than or equal to 32% in most of these northern regions. These remaining northward populations are patchily distributed, but occasionally harbour high densities of larger *Pycnopodia*. As with the rate of disease spread, the drivers of this variability could lie with the host, the disease or environmental interactions between the two. However, the variation in mortality, particularly within the northern regions, creates an excellent opportunity for future research.

The 4.5-fold increase in the importance of temperature in predicting *Pycnopodia* distribution post-outbreak suggests temperature could be a driving force behind the observed latitudinal patterns in the speed and severity of the disease. After SSWD, the relationship between *Pycnopodia* occurrence and temperature became strongly negative from 5 to 20°C, suggesting a

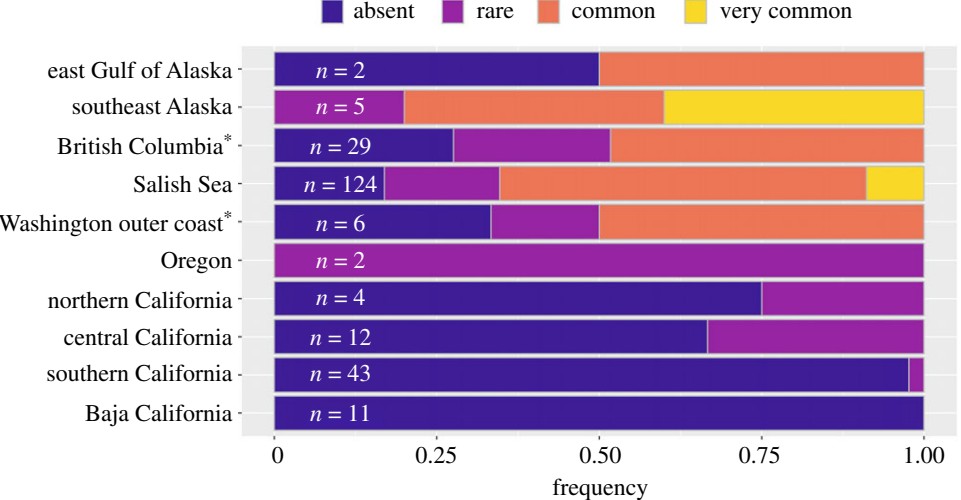

**Figure 5.** The frequency with which *Pycnopodia helianthoides* remnant populations were observed from 2017 to 2020 in each region. Surveys were aggregated into 16 km$^2$ grid cells and grid cells were only included if they contained shallow (less than 25 m) surveys from at least three different years from 2017 to 2020. $n$ = the number of grid cells that fit this criterion ($n = 0$ for Aleutians and west GOA; not shown). Each grid cell was classified by the per cent of total surveys that observed *Pycnopodia*: absent = 0%, rare = less than 25%, common = less than 90% and very common ≥90%. Asterisk: British Columbia and Washington outer coast exclude the Salish Sea. (Online version in colour.)

disease-mediated shift in temperature associations. This is consistent with experimental studies that have shown warmer temperatures cause SSWD to progress more quickly and increase sea star mortality [16–18]. These studies documented increased individual-level impacts of SSWD over a range of 9–19°C, which mirrors the decreasing incidence of *Pycnopodia* over this range of temperatures currently.

Across systems, elevated temperatures generally increase virulence, growth rates and overwintering success of many pathogens, and heat stress in host organisms shifts energy allocation towards metabolic demands, leaving fewer resources for immunological functions [36,37]. Thus, the putative link between temperature and SSWD speed and severity is unsurprising. While we infer that temperature drove the latitudinal patterns documented here, this association is correlational and does not rule out confounding variables of temperature such as latitude, coastline complexity or nutrients (electronic supplementary material, figure S5). For instance, an alternative hypothesis for the geographic patterns seen here is that if a latitudinal gradient exists in genetic resistance to SSWD, with greater resistance in the northern half of the range than the south, then this could have created the same pattern in SSWD impacts that we infer temperature did. Additionally, this continental-scale analysis glosses over important regional-scale variability, and regional to local-scale investigations of the relationship between abiotic variables and population-level resistance to SSWD are warranted. While our analysis is strongly suggestive, it is not conclusive.

Additionally, whether climate change or warm temperatures triggered the outbreak remains unknown. Harvell *et al.* [6] showed that warm temperature anomalies explained more than a third of the variance in *Pycnopodia* outbreak timing in the Salish Sea [6]. Furthermore, Aalto *et al.* [19] modelled the initial outbreak spread dynamics and suggested that warm temperatures can trigger disease and increase mortality [19]. Conversely, several studies found that warmer ocean temperatures were not associated with SSWD outbreak timing in *Pisaster ochraceus* in Oregon and California [8,21]. Though we lack a mechanistic understanding of whether

temperature or climate change triggered the SSWD outbreak, this study adds to existing evidence that the speed and severity of SSWD are greater in warmer waters.

A recent hypothesis advanced from laboratory experiments suggests that elevated dissolved organic matter or low-dissolved oxygen triggers SSWD [15]. Because continental scale, near shore estimates of these variables do not exist at high enough spatial resolution to be incorporated into our models, we were unable to test this hypothesis. However, to our knowledge, no large-scale hypoxic event occurred prior to the SSWD epidemic. Further, large-scale hypoxic events have occurred periodically in places like Oregon [38] in recent decades with no subsequent outbreaks of SSWD. The proposed link between elevated dissolved organic matter, low-dissolved oxygen and SSWD remains a hypothesis that requires further evaluation in the field.

## (b) Supporting recovery

We found little evidence of region-wide recovery in *Pycnopodia* since 2017, and many southern regions show evidence of functional extirpation. Although we are aware of recent juvenile recruitment events in the GOA, southeast Alaska and British Columbia (K. Gavenus & P. Raimondi 2021, personal communication; A.L.G. 2017–2021, personal observation), in British Columbia juveniles appear to be failing to grow into adults, presumably because of recurring outbreaks of SSWD (A.L.G. 2017–2021, personal observation). Spatial variability in the impacts of SSWD creates variable recovery pathways for *Pycnopodia*. For example, protecting surviving adults in more northern regions will likely be critical for natural recovery. While *Pycnopodia* are not targeted in fisheries, adults may be killed as bycatch in trap and trawl fisheries, (T. Frierson 2021, personal communication) and bycatch mortality should be considered in recovery planning.

Southward, natural recovery will probably be impeded by low larval availability and Allee effects. We believe the time has come for active recovery of this IUCN-listed Critically Endangered species in the southern half of its range

[11]. Active recovery strategies include captive breeding plus reintroduction of young animals and translocations of adult animals from extant to locally extinct areas. The recent investment shows that captive breeding is feasible, but the capacity and effort required to scale breeding programmes to support recovery over large areas requires further investigation (J. Hodin 2021, personal communication). Recent work by Schiebelhut et al. [39] suggests a genetic underpinning for SSWD resistance, so it may be advisable to selectively breed resistant adults or to reintroduce a high number of younger, smaller and genetically diverse animals [39]. Comparatively, translocations are lower cost compared to captive rearing. However, translocation is problematic due to a lack of robust donor populations, the logistics of crossing international borders, losses of re-introduced animals to SSWD in transplanted locations, and risks of SSWD and other unintended introductions into target areas.

Closing key research gaps will increase the capacity for recovering Pycnopodia populations. Research into the aetiology of SSWD, how disease susceptibility varies among individuals, life stages and populations, and how environmental factors influence susceptibility and resistance are crucial. We also lack a basic understanding of important life-history information for Pycnopodia, including reproductive phenology, growth rates and genetic structure. Finally, while multiple studies have found that Pycnopodia can reduce grazing by sea urchins in subtidal kelp forests, we lack information on the variability in the magnitude and spatial scale of this interaction across Pycnopodia's range [10,12,13]. Understanding the ecological, economic and social impacts of Pycnopodia recovery as a tool for restoring degraded kelp forest ecosystems is urgently needed given recent collapses in kelp forests within its range [34].

In times of rapidly changing ocean conditions, the plight of Pycnopodia highlights the importance of enhancing long-term monitoring (LTM) programmes to allow us to better monitor, maintain and strengthen the resilience of marine ecosystems. We cannot overstate the importance of well-coordinated LTM to this effort and future MME work. The 'what' and 'how' of LTM is also key. For example, if size frequency and vital rates data were available for Pycnopodia, size-based population models could have been constructed to help assess population growth rates and project time to quasi-extinction. We see a need to add information on organism size frequency, health, genetic diversity and ecological interactions to the ongoing LTM of population incidence and density. Additionally, citizen science, a crucial component of this study, increases the spatial scale and frequency of LTM and increases the likelihood of detecting incipient MMEs. For wide-ranging marine species, cross-boundary coordination of consistent minimum monitoring standards and data sharing pathways are critical. Overall, remarkable circumstances call for remarkable investment in and development of broad-scale LTM programmes.

## 5. Conclusion

This study documents the disease-driven extirpation of a marine predator over 2700 km of coastline. Eight years after the SSWD outbreak began, the causative agent(s) of the disease remain unknown. This mismatch between the severity of the epidemic and the state of knowledge highlights the paucity of tools and support available to understand and respond to disease-driven MMEs, particularly in species that are neither commercially important nor charismatic. Currently, very few management, conservation or policy efforts have been developed to respond to MMEs in marine wildlife. Science, funding, management, conservation and policy often move slowly, yet if the frequency of MMEs continues to increase, institutions will need to respond much more quickly than they have to the SSWD epidemic. Increasing the capacity to monitor a wide variety of species, detect early warning signs of MMEs and rapidly research and respond to them will be increasingly important in the coming years.

Data accessibility. The compiled dataset and code to replicate the analyses conducted and figures created for this paper are available from the Dryad Digital Repository: https://doi.org/10.5061/dryad.9kd51c5hg [40].

Authors' contributions. S.L.H.: conceptualization, data curation, formal analysis, funding acquisition, investigation, methodology, project administration, visualization, writing-original draft, writing-review and editing; V.R.S.: formal analysis, investigation, methodology, visualization, writing-original draft, writing-review and editing; W.N.H.: conceptualization, funding acquisition, writing-original draft, writing-review and editing; A.L.G.: formal analysis, investigation, methodology, writing-original draft, writing-review and editing; S.I.L.: formal analysis, methodology, visualization, writing-original draft, writing-review and editing; R.B.-L.: methodology, writing-original draft, writing-review and editing; F.T.F.: methodology, writing-original draft, writing-review and editing; L.L.: methodology, writing-original draft, writing-review and editing; L.R.-B.: methodology, writing-original draft, writing-review and editing; A.K.S.: methodology, writing-original draft, writing-review and editing; S.A.G.: conceptualization, data curation, formal analysis, funding acquisition, investigation, methodology, project administration, supervision, visualization, writing-original draft, writing-review and editing.

All authors gave final approval for publication and agreed to be held accountable for the work performed therein.

Competing interests. We declare no competing interests.

Funding. This work was supported by the Nature Conservancy and a National Science Foundation Graduate Research Fellowship.

Acknowledgements. We thank Lindsey Aylesworth, Tristan Blaine, Jenn Burt, Mark Carr, Henry Carson, Jenn Caselle, Ryan Cloutier, Isabelle Côté, Tom Dean, Eduardo Diaz, David Duggins, George Esslinger, Jan Freiwald, Alejandro Frid, Taylor Frierson, Rani Gaddam, Katie Gavenus, Donna Gibbs, the Haida Nation, the Heiltsuk Nation, Chris Jenkins, Cori Kane, Aimie Keller, the Kitasoo/Xai'xais Nation, Brenda Konar, Kristy Kroeker, Andy Lauermann, Julio Lorda, Dan Malone, Scott Marion, Dan McNeill, Fiorenza Micheli, Melissa Miner, Gaby Montaño, the Nuxalk Nation, Dan Okamoto, Christy Pattengill-Semmens, Mike Prall, Pete Raimondi, Nancy Roberson, Dirk Rosen, Jessica Schultz, Ole Shelton, Jorge Torre, Guillermo Torres-Moye, Jane Watson, Ben Weitzman, Greg Williams and the Wuikinuxv Nation and their associated institutions (electronic supplementary material table S1) for their willingness to share data for this effort. We thank Norah Eddy, Joe Gaydos, Drew Harvell, Jason Hodin, Erin Meyer, Kirsten Alvstad and Josh Havelind for their help and guidance. Please see the electronic supplementary material for a full list of acknowledgements. The scientific results and conclusions, as well as any views or opinions expressed herein, are those of the authors and do not necessarily reflect the views of NOAA or the Department of Commerce.

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
