## [Peer Review File · Proceedings of the Royal Society B: Biological Sciences]

Review History

RSPB-2021-1195.R0 (Original submission)

Review form: Reviewer 1

Recommendation

Major revision is needed (please make suggestions in comments)

Scientific importance: Is the manuscript an original and important contribution to its field?

Excellent

General interest: Is the paper of sufficient general interest?

Good

Quality of the paper: Is the overall quality of the paper suitable?

Good

Is the length of the paper justified?

Yes

Should the paper be seen by a specialist statistical reviewer?

No

Do you have any concerns about statistical analyses in this paper? If so, please specify them explicitly in your report.

Yes

It is a condition of publication that authors make their supporting data, code and materials available - either as supplementary material or hosted in an external repository. Please rate, if applicable, the supporting data on the following criteria.

Is it accessible?

Yes

Is it clear?

Yes

Is it adequate?

Yes

Do you have any ethical concerns with this paper?

No

Comments to the Author

The manuscript characterizes the time frame of the Sea Star Wasting epidemic for the ecologically important sea star *Pycnopodia helianthoides*. The authors draw upon a wealth of survey data to estimate the timing of decline and compare the pre- and post-disease distribution with environmental conditions to identify potential factors triggering decline. The study documents a lack of recovery for this species in part of its range. The implications of this massive decline include ecological changes and a potential preview of future mass mortality events in echinoderms. Overall, the paper is an important contribution to a continental-scale understanding of the progression of Sea Star Wasting Disease, about which little was known before the current epidemic. This high-profile disease and its potential environmental causes is of interest to a broad readership.

The manuscript is well-structured and up front about the limitations of using this collection of survey data to elucidate epidemic timing. The methods chosen are appropriate, particularly given the challenges of documenting a rapidly progressing disease over a vast geographic extent. However, the methods section would benefit from clearer explanations of why specific threshold levels were chosen. Additionally, the Maxent analysis linking pre- and post-epidemic environmental conditions to species distribution would benefit from two changes: (1) instead of using a 30-year climatology for the post-disease species distribution model, using the climatology data from the post-disease period would be more appropriate. (2) The vast amount of environmental variation encountered from Baja to Alaska could be swamping potential signal from environmental patterns. The authors conclude that temperature may be linked to the resulting sea star declines. It is difficult to know whether this association is overstated without understanding whether this is also a hallmark of sea star declines within regions. Performing Maxent on each standalone region in addition to what has already been done on a whole-coast scale would strengthen the analysis and better clarify the role of temperature versus other regional environmental and ecological differences.

Comments:

Line 25. "We are left dramatically unprepared" is a departure from the focus of the rest of the paragraph. I suggest rephrasing.

Line 52. In question #3 you ask how were the environmental variables associated with *Pycnopodia* distribution different before and after the outbreak instead of asking what conditions

were associated with the timing of the declines that you are documenting, and I think many readers are going to wonder why you didn't ask the latter question. The points you make at the in the prior paragraph hint at this, but the introduction could use some language that pulls together why thus far it has been difficult to link environmental conditions directly to the timing of decline.

Line 56-58. This is an impressive compilation of data.

Line 85. Why did you choose the 10th percentile as the cutoff for the phases? Please clarify the rationale.

Line 90. Similar to above, why did you choose a 75% decline as the threshold for the population crash? Is this rooted in the species' population dynamics or ability to perform its function as a predator?

Line 127. Briefly list which variables were highly correlated and removed. How did you choose which variable to eliminate? For example, 90th percentile temperature is likely highly correlated with latitude.

Line 128. For environmental factors that change over time (e.g., temperature, chlorophyll), it would be useful to run the Maxent model using separate climatologies from the 2017-2020 period to evaluate the abiotic factors associated with the remaining population distribution. For example, if temperatures have increased substantially, using the 30-year mean temperature is no longer a reflection of the conditions Pycnopodia must deal with.

Line 133. Running the Maxent model separately for each region in addition to the whole coast would help to account for the regional differences across this large geographic range. If the same abiotic variables emerged as important in most of the regions, it would increase confidence in what factors are truly associated with the shifts in distribution and what is merely a product of the major environmental and ecological variation between regions.

Line 168. Why is your threshold for 'Rare' <30%, when your threshold for a crashing population was 25%? It would be more logical of thresholds of rarity and a crashing population were linked.

Line 204. "Still severe" would be a better descriptor than "somewhat less severe" in this case.

Line 232-233. Does using the 2017 - 2020 mean temperature instead of the 30-year mean influence this result, or does running a separate Maxent model for each region influence this result?

Line 275-287. Why does knowing the regional differences in timelines of the population crash matter? What do we gain from understanding it?

Line 289. Is there some way to separate out influence of temperature and latitude in the Maxent model? It is difficult to tell what role temperature versus latitude had in the shifts in occurrence probability. If a spatial pattern emerged where hotspots within a region had higher Pycnopodia decline than the area around it, that would be very compelling evidence. To borrow an example from a different sea star species, at this coast-wide scale, there was low evidence for temperature being associated with *Pisaster's* decline, even though lab experiments showed that high temperatures hasten the course of SSWD.

Fig 3. Putting the data range under the panel title would improve this figure.

Supplemental Material

Line 230. There appears to be an empty table above table 4.

Fig. S2. Are the arrows pointing to the epidemic phase timeline supposed to line up with the shifts in phase? It seems odd that they don't.

Fig. S3. With the major differences in densities between the regions, this figure would be better split up into a series of panels so that the trends in each region can be examined. You can group them by larger geographic areas such as quarters of their range or individual regions.

Review form: Reviewer 2

Recommendation

Accept with minor revision (please list in comments)

Scientific importance: Is the manuscript an original and important contribution to its field?

Excellent

General interest: Is the paper of sufficient general interest?

Excellent

Quality of the paper: Is the overall quality of the paper suitable?

Excellent

Is the length of the paper justified?

Yes

Should the paper be seen by a specialist statistical reviewer?

Yes

Do you have any concerns about statistical analyses in this paper? If so, please specify them explicitly in your report.

No

It is a condition of publication that authors make their supporting data, code and materials available - either as supplementary material or hosted in an external repository. Please rate, if applicable, the supporting data on the following criteria.

Is it accessible?

Yes

Is it clear?

Yes

Is it adequate?

Yes

Do you have any ethical concerns with this paper?

No

Comments to the Author

This manuscript compiles range-wide survey data on Pycnopodia to describe the recent progression of SSWD and population trends in the species. This compilation of data is striking and impactful all by itself. The authors also use species distribution models (SDMs) to generate hypotheses on potential environmental drivers of SSWD in this species. I will note that I am not expert in species distribution modeling, so I hope that another reviewer with that specific

expertise evaluates that component of the manuscript.

I really have no major criticisms of the manuscript. The dataset is unique in scale, the analyses all seem appropriate, and the writing is clear and concise. I thank the authors for pulling all these data together; this is an important paper.

I do have a few minor suggestions for improvement, however, listed below.

1) The authors used data from both *Pisaster ochraceus* and *Pycnopodia* to identify the date SSWD was first observed (in *Pycnopodia*), noting in the methods that they did so because there are more observations for *Pisaster*, and because inspection of the data showed co-incident outbreaks between the two species. That second point is important, because the two species likely differ in distribution, with *Pisaster* being much more common in shallower water (and especially the intertidal zone), and *Pycnopodia* in the shallow and deeper subtidal. Given the potential importance of temperature in influencing SSWD, and systematic differences in temperature with depth, this might lead to some bias in the estimation of date SSWD was first observed, which would affect estimates of emerging epidemic and outbreak duration. I would suggest that the authors mention this possibility briefly in the manuscript, and provide at least some evidence in support of the “co-incident outbreaks between the two species” in the supplementary material.

2) The authors treat the recent epidemic, which started in 2013, as unique (e.g., in the introduction, “The SSWD epidemic... began in 2013.” But there have been earlier outbreaks of disease among west coast seastars, including *Pycnopodia*, that are indistinguishable in symptoms from those of the current epidemic. E.g., see Eckert et al. 1999, Sea star disease and population declines at the Channel Islands. Proc 5th CA Channel Islands Symp. 1999;5: 390-393 (pdf freely available online). We don't know the etiological agent(s) of the current or prior epidemics, so cannot say that the causes are the same. However, it would be nice to note somewhere in the introduction or discussion that the current epidemic is not necessarily a unique event in the history of *Pycnopodia* or west coast seastars more generally (though it may be greater in intensity and spatial extent than prior epidemics).

One other related issue is that prior epidemics (e.g., 1982-1984, and 1997-1998) may have had some effect on historic population sizes. Such potential effects are not visible from the data presented, because though the authors collected survey data spanning 1967 to 2020, they only illustrate (in Fig. 1a) predicted incidence from 2012 to 2020. I don't know why this is, though perhaps the older data are too patchy in space to generate predictions for all regions? If it is possible to generate predicted incidence all the way back to the late 1960s, that would be very interesting for readers to see, and would provide some very useful historical information on variation in population size and perhaps the timing of prior epidemics in this species.

3) A very minor point - Fig. 1b is cited in the results before Fig. 1a. Why not simply switch those two so you can cite figures in order? I suspect this was a minor error introduced in revision, because in the second paragraph of the discussion the authors refer to Fig 1b when I think they mean to refer to the current Fig. 1a (which really should be b). The current Fig. 1b does not illustrate the point they are trying to make in this paragraph.

Decision letter (RSPB-2021-1195.R0)

28-Jun-2021

Dear Ms Hamilton:

Your manuscript has now been peer reviewed and the reviews have been assessed by an Associate Editor. The reviewers' comments (not including confidential comments to the Editor) and the comments from the Associate Editor are included at the end of this email for your reference. As you will see, the reviewers and the Editors have raised some concerns with your manuscript and we would like to invite you to revise your manuscript to address them.

Research ethics:

Use of animals and field studies:

It is a condition of publication that you make available the data and research materials supporting the results in the article. Please see our Data Sharing Policies (<https://royalsociety.org/journals/authors/author-guidelines/#data>). Datasets should be deposited in an appropriate publicly available repository and details of the associated accession number, link or DOI to the datasets must be included in the Data Accessibility section of the article (<https://royalsociety.org/journals/ethics-policies/data-sharing-mining/>). Reference(s) to datasets should also be included in the reference list of the article with DOIs (where available).

Please submit a copy of your revised paper within three weeks. If we do not hear from you within this time your manuscript will be rejected. If you are unable to meet this deadline please let us know as soon as possible, as we may be able to grant a short extension.

Best wishes,
Dr Daniel Costa
mailto:proceedingsb@royalsociety.org

Associate Editor

Board Member: 1

Comments to Author:

Both reviewers have identified that the manuscript is an important contribution to a continental-scale understanding of the progression of Sea Star Wasting Disease. Both have provided suggestions for how the manuscript could be improved to bring it to a publishable quality, in terms of clarifying content but also in considering the spatial habitat modelling undertaken. I invite the authors to carefully consider both reviews and carefully consider how they might address the concerns raised in regards to better defining drivers associated with environmental variability.

Reviewer(s)' Comments to Author:

Referee: 1

Comments to the Author(s)

The manuscript characterizes the time frame of the Sea Star Wasting epidemic for the ecologically important sea star *Pycnopodia helianthoides*. The authors draw upon a wealth of survey data to estimate the timing of decline and compare the pre- and post-disease distribution with environmental conditions to identify potential factors triggering decline. The study documents a lack of recovery for this species in part of its range. The implications of this massive decline include ecological changes and a potential preview of future mass mortality events in echinoderms. Overall, the paper is an important contribution to a continental-scale understanding of the progression of Sea Star Wasting Disease, about which little was known

before the current epidemic. This high-profile disease and its potential environmental causes is of interest to a broad readership.

The manuscript is well-structured and up front about the limitations of using this collection of survey data to elucidate epidemic timing. The methods chosen are appropriate, particularly given the challenges of documenting a rapidly progressing disease over a vast geographic extent. However, the methods section would benefit from clearer explanations of why specific threshold levels were chosen. Additionally, the Maxent analysis linking pre- and post-epidemic environmental conditions to species distribution would benefit from two changes: (1) instead of using a 30-year climatology for the post-disease species distribution model, using the climatology data from the post-disease period would be more appropriate. (2) The vast amount of environmental variation encountered from Baja to Alaska could be swamping potential signal from environmental patterns. The authors conclude that temperature may be linked to the resulting sea star declines. It is difficult to know whether this association is overstated without understanding whether this is also a hallmark of sea star declines within regions. Performing Maxent on each standalone region in addition to what has already been done on a whole-coast scale would strengthen the analysis and better clarify the role of temperature versus other regional environmental and ecological differences.

Comments:

Line 25. "We are left dramatically unprepared" is a departure from the focus of the rest of the paragraph. I suggest rephrasing.

Line 52. In question #3 you ask how were the environmental variables associated with Pycnopia distribution different before and after the outbreak instead of asking what conditions were associated with the timing of the declines that you are documenting, and I think many readers are going to wonder why you didn't ask the latter question. The points you make at the in the prior paragraph hint at this, but the introduction could use some language that pulls together why thus far it has been difficult to link environmental conditions directly to the timing of decline.

Line 56-58. This is an impressive compilation of data.

Line 85. Why did you choose the 10th percentile as the cutoff for the phases? Please clarify the rationale.

Line 90. Similar to above, why did you choose a 75% decline as the threshold for the population crash? Is this rooted in the species' population dynamics or ability to perform its function as a predator?

Line 127. Briefly list which variables were highly correlated and removed. How did you choose which variable to eliminate? For example, 90th percentile temperature is likely highly correlated with latitude.

Line 128. For environmental factors that change over time (e.g., temperature, chlorophyll), it would be useful to run the Maxent model using separate climatologies from the 2017-2020 period to evaluate the abiotic factors associated with the remaining population distribution. For example, if temperatures have increased substantially, using the 30-year mean temperature is no longer a reflection of the conditions Pycnopia must deal with.

Line 133. Running the Maxent model separately for each region in addition to the whole coast would help to account for the regional differences across this large geographic range. If the same abiotic variables emerged as important in most of the regions, it would increase confidence in what factors are truly associated with the shifts in distribution and what is merely a product of the major environmental and ecological variation between regions.

Line 168. Why is your threshold for 'Rare' <30%, when your threshold for a crashing population was 25%? It would be more logical if thresholds of rarity and a crashing population were linked.

Line 204. "Still severe" would be a better descriptor than "somewhat less severe" in this case.

Line 232-233. Does using the 2017 - 2020 mean temperature instead of the 30-year mean influence this result, or does running a separate Maxent model for each region influence this result?

Line 275-287. Why does knowing the regional differences in timelines of the population crash matter? What do we gain from understanding it?

Line 289. Is there some way to separate out influence of temperature and latitude in the Maxent model? It is difficult to tell what role temperature versus latitude had in the shifts in occurrence probability. If a spatial pattern emerged where hotspots within a region had higher Pycnopodia decline than the area around it, that would be very compelling evidence. To borrow an example from a different sea star species, at this coast-wide scale, there was low evidence for temperature being associated with *Pisaster's* decline, even though lab experiments showed that high temperatures hasten the course of SSWD.

Fig 3. Putting the data range under the panel title would improve this figure.

Supplemental Material

Line 230. There appears to be an empty table above table 4.

Fig. S2. Are the arrows pointing to the epidemic phase timeline supposed to line up with the shifts in phase? It seems odd that they don't.

Fig. S3. With the major differences in densities between the regions, this figure would be better split up into a series of panels so that the trends in each region can be examined. You can group them by larger geographic areas such as quarters of their range or individual regions.

Referee: 2

Comments to the Author(s)

This manuscript compiles range-wide survey data on Pycnopodia to describe the recent progression of SSWD and population trends in the species. This compilation of data is striking and impactful all by itself. The authors also use species distribution models (SDMs) to generate hypotheses on potential environmental drivers of SSWD in this species. I will note that I am not expert in species distribution modeling, so I hope that another reviewer with that specific expertise evaluates that component of the manuscript.

I really have no major criticisms of the manuscript. The dataset is unique in scale, the analyses all seem appropriate, and the writing is clear and concise. I thank the authors for pulling all these data together; this is an important paper.

I do have a few minor suggestions for improvement, however, listed below.

1) The authors used data from both *Pisaster ochraceus* and Pycnopodia to identify the date SSWD was first observed (in Pycnopodia), noting in the methods that they did so because there are more observations for *Pisaster*, and because inspection of the data showed co-incident outbreaks between the two species. That second point is important, because the two species likely differ in distribution, with *Pisaster* being much more common in shallower water (and especially the intertidal zone), and Pycnopodia in the shallow and deeper subtidal. Given the potential importance of temperature in influencing SSWD, and systematic differences in temperature with depth, this might lead to some bias in the estimation of date SSWD was first observed, which

would affect estimates of emerging epidemic and outbreak duration. I would suggest that the authors mention this possibility briefly in the manuscript, and provide at least some evidence in support of the “co-incident outbreaks between the two species” in the supplementary material.

2) The authors treat the recent epidemic, which started in 2013, as unique (e.g., in the introduction, “The SSWD epidemic... began in 2013.” But there have been earlier outbreaks of disease among west coast seastars, including Pycnopodia, that are indistinguishable in symptoms from those of the current epidemic. E.g., see Eckert et al. 1999, Sea star disease and population declines at the Channel Islands. Proc 5th CA Channel Islands Symp. 1999;5: 390-393 (pdf freely available online). We don't know the etiological agent(s) of the current or prior epidemics, so cannot say that the causes are the same. However, it would be nice to note somewhere in the introduction or discussion that the current epidemic is not necessarily a unique event in the history of Pycnopodia or west coast seastars more generally (though it may be greater in intensity and spatial extent than prior epidemics).

One other related issue is that prior epidemics (e.g., 1982-1984, and 1997-1998) may have had some effect on historic population sizes. Such potential effects are not visible from the data presented, because though the authors collected survey data spanning 1967 to 2020, they only illustrate (in Fig. 1a) predicted incidence from 2012 to 2020. I don't know why this is, though perhaps the older data are too patchy in space to generate predictions for all regions? If it is possible to generate predicted incidence all the way back to the late 1960s, that would be very interesting for readers to see, and would provide some very useful historical information on variation in population size and perhaps the timing of prior epidemics in this species.

3) A very minor point - Fig. 1b is cited in the results before Fig. 1a. Why not simply switch those two so you can cite figures in order? I suspect this was a minor error introduced in revision, because in the second paragraph of the discussion the authors refer to Fig 1b when I think they mean to refer to the current Fig. 1a (which really should be b). The current Fig. 1b does not illustrate the point they are trying to make in this paragraph.

Author's Response to Decision Letter for (RSPB-2021-1195.R0)

See Appendix A.

Decision letter (RSPB-2021-1195.R1)

27-Jul-2021

Dear Ms Hamilton

I am pleased to inform you that your Review manuscript RSPB-2021-1195.R1 entitled "Disease-driven mass mortality event leads to widespread extirpation and variable recovery potential of a marine predator across the eastern Pacific" has been accepted for publication in Proceedings B.

The referee(s) do not recommend any further changes. Therefore, please proof-read your manuscript carefully and upload your final files for publication. Because the schedule for publication is very tight, it is a condition of publication that you submit the revised version of your manuscript within 7 days. If you do not think you will be able to meet this date please let me know immediately.

To upload your manuscript, log into <http://mc.manuscriptcentral.com/prsb> and enter your Author Centre, where you will find your manuscript title listed under "Manuscripts with Decisions." Under "Actions," click on "Create a Revision." Your manuscript number has been appended to denote a revision.

You will be unable to make your revisions on the originally submitted version of the manuscript. Instead, upload a new version through your Author Centre.

1) A text file of the manuscript (doc, txt, rtf or tex), including the references, tables (including captions) and figure captions. Please remove any tracked changes from the text before submission. PDF files are not an accepted format for the "Main Document".

2) A separate electronic file of each figure (tiff, EPS or print-quality PDF preferred). The format should be produced directly from original creation package, or original software format. Please note that PowerPoint files are not accepted.

3) Electronic supplementary material: this should be contained in a separate file from the main text and the file name should contain the author's name and journal name, e.g. `authorname_procb_ESM_figures.pdf`

All supplementary materials accompanying an accepted article will be treated as in their final form. They will be published alongside the paper on the journal website and posted on the online figshare repository. Files on figshare will be made available approximately one week before the accompanying article so that the supplementary material can be attributed a unique DOI. Please see: <https://royalsociety.org/journals/authors/author-guidelines/>

4) Data-Sharing and data citation

It is a condition of publication that data supporting your paper are made available. Data should be made available either in the electronic supplementary material or through an appropriate repository. Details of how to access data should be included in your paper. Please see <https://royalsociety.org/journals/ethics-policies/data-sharing-mining/> for more details.

<http://datadryad.org/submit?journalID=RSPB&manu=RSPB-2021-1195.R1> which will take you to your unique entry in the Dryad repository.

Once again, thank you for submitting your manuscript to Proceedings B and I look forward to receiving your final version. If you have any questions at all, please do not hesitate to get in touch.

Sincerely,

Dr Daniel Costa

Associate Editor Board Member

Comments to Author:

Overall, the authors have done a good job in addressing the reviewers concerns and revising their manuscript. I just have one minor comment that the authors that the authors should consider:

The introduction at present appears to mostly focus on Pycnopodia and yet the analyses focuses on both Pisaster and Pycnopodia with the majority of records used in the analyses derived from Pisaster. Some recognition of Pisaster and the relevance of this species in terms of the wasting disease in the introduction, or alternatively some clearer clarification on how the observations of Pisaster were representative of Pycnopodia (and therefore could be used as a proxy) in the Methods would be of use to the reader. This should only require the addition of a couple of sentences to either section of the manuscript.

Decision letter (RSPB-2021-1195.R2)

04-Aug-2021

Dear Ms Hamilton

I am pleased to inform you that your manuscript entitled "Disease-driven mass mortality event leads to widespread extirpation and variable recovery potential of a marine predator across the eastern Pacific" has been accepted for publication in Proceedings B.

Data Accessibility section

Open Access

Paper charges

Sincerely,
Editor, Proceedings B
mailto: proceedingsb@royalsociety.org

Appendix A

Associate Editor

Board Member: 1

Comments to Author:

Both reviewers have identified that the manuscript is an important contribution to a continental-scale understanding of the progression of Sea Star Wasting Disease. Both have provided suggestions for how the manuscript could be improved to bring it to a publishable quality, in terms of clarifying content but also in considering the spatial habitat modelling undertaken. I invite the authors to carefully consider both reviews and carefully consider how they might address the concerns raised in regards to better defining drivers associated with environmental variability.

Reviewer(s)' Comments to Author:

Referee: 1

Comments to the Author(s)

The manuscript characterizes the time frame of the Sea Star Wasting epidemic for the ecologically important sea star *Pycnopodia helianthoides*. The authors draw upon a wealth of survey data to estimate the timing of decline and compare the pre- and post-disease distribution with environmental conditions to identify potential factors triggering decline. The study documents a lack of recovery for this species in part of its range. The implications of this massive decline include ecological changes and a potential preview of future mass mortality events in echinoderms. Overall, the paper is an important contribution to a continental-scale understanding of the progression of Sea Star Wasting Disease, about which little was known before the current epidemic. This high-profile disease and its potential environmental causes is of interest to a broad readership.

The manuscript is well-structured and up front about the limitations of using this collection of survey data to elucidate epidemic timing. The methods chosen are appropriate, particularly given the challenges of documenting a rapidly progressing disease over a vast geographic extent. However, the methods section would benefit from clearer explanations of why specific threshold levels were chosen. Additionally, the Maxent analysis linking pre- and post-epidemic environmental conditions to species distribution would benefit from two changes: (1) instead of using a 30-year climatology for the post-disease species distribution model, using the climatology data from the post-disease period would be more appropriate. (2) The vast amount of environmental variation encountered from Baja to Alaska could be swamping potential signal from environmental patterns. The authors conclude that temperature may be linked to the resulting sea star declines. It is difficult to know whether this association is overstated without understanding whether this is also a hallmark of sea star declines within regions. Performing Maxent on each standalone region in addition to what has already been done on a whole-coast scale would strengthen the analysis and better clarify the role of temperature versus other regional environmental and ecological differences.

We appreciate this useful feedback from the editor and the reviewers. The major concerns of Reviewer 1 regard 1) why specific thresholds were chosen, 2) using long-term climatologies vs. time period specific environmental data, and 3) using regional Maxent models to more deeply investigate the role of environmental variables on disease progression.

Regarding concern 1, we have added language in both the manuscript and the supplement that explain our use of various thresholds. In most of the cases highlighted in the line comments, these thresholds were chosen because they were biologically and mathematically realistic. However, we acknowledge that the exact number (i.e. 10% vs. 15%) was not supported by precise

biological information and have addressed this point. Please see individual line comments for more details

We have two responses regarding concern 2. First, this was partially an oversight on our part in the Methods section, and we thank the reviewer for their careful reading. To be clear, we did indeed calculate mean chlorophyll and 90th percentile of temperature for the 2009-2012 period for the pre-outbreak model and for the 2017-2020 period for the current model. This was possible due to the availability of high-resolution, gridded, 8-day raster datasets of these variables derived from NASA's MODIS Aqua sensor on Google Earth Engine. We have now added this information to the manuscript. It is unlikely that depth changed substantially at large spatial scales between the two periods so that was not recalculated for the two different periods.

We agree with the reviewer that ideally we would have calculated mean salinity and substrate type for both the 2009-2012 and the 2017-2020 period. Unfortunately, neither variable is easily mapped from remote sensing. These variables are usually mapped using in situ depth profiles run from vessels. For instance, the highest resolution salinity remote sensing data that we are aware of is the NASA Aquarius sensor that maps salinity at a 1-degree latitude resolution, far too coarse for this study. Thus, the data available for these variables is neither regular enough nor fine-scale enough to map these variables across the entire continent for each time period. For both datasets, the best maps available with the needed resolution (11 km grids) were only available as long-term estimates. Making these maps were entire projects on their own (salinity - <https://www.sciencedirect.com/science/article/pii/S0079661114000160?via%3Dihub>, substrate - <https://www.sciencedirect.com/science/article/pii/S0278434307003159?via%3Dihub>), and creating our own set for the specified time periods is outside the scope of this paper.

Overall, despite the shortcomings of long term climatologies, we believe that these climatologies do represent continental-scale patterns in substrate and salinity. For instance, in this region, salinity is most strongly influenced by the addition of freshwater from large rivers in the northern part of the range. While anthropogenic impacts to these rivers are likely impacting ocean salinity, it is unlikely that the patterns associated with these major rivers have disappeared between the early 2000s and the late 2010s. To test this and verify our use of long-term climatologies, we investigated how using a long-term climatology of temperature vs. 2009-2012/2017-2020 temperature data impacted the models, and found that neither the pre-outbreak nor the current model was sensitive to this change. We have added language to the Supplement documenting this limitation.

Finally, we have extensively considered Reviewer 1's concern 3. We agree that region-level species distribution models would strengthen the conclusions from our continental-scale SDMs. However, we have several reservations with this suggestion including:

- 1. The spatial scale of our environmental data - the environmental variables used in these models (depth, substrate, 90th percentile of temp., mean chlorophyll, and mean salinity) are in 11km grid raster datasets. While this captures substantial environmental variability at the continental scale, it becomes problematic at the regional scale, particularly in regions with complex coastlines such as the Salish Sea or British Columbia. For instance, even along Oregon's relatively linear coastline Hemery et al. (2016) conducted their regional sea star Maxent models at a 2km grid resolution. We feel that using environmental data with the 11km resolution would produce inadequate regional models, and to create truly useful regional-level models, we would need to find new environmental data that may not exist at the required resolution. To do this would be a substantial undertaking that would be intensive enough to represent its own paper.*
- 2. Coverage of Pycnophodia sightings - Maxent SDMs rely on observations of the species in question to build the model. By compiling data from 30+ organizations over 3 countries*

and 3 decades, we were able to create a dataset that had enough Pycnopodia observations to create powerful Maxent SDMs. We chose to do a continental-scale analysis because it leveraged the sheer amount of data we had on Pycnopodia sightings even though there were holes in the coverage in both space and time. However, as you begin to zoom in to individual regions, many do not have enough data on Pycnopodia sightings to create meaningful, localized pre-outbreak and current SDMs. For instance, in our dataset, Baja California, southern California, central California, northern California, Oregon, and Washington outer coast all have recorded only a few Pycnopodia in the 2017-2020 period. Conversely, in the northern half of the range where fewer surveys are conducted, there are limited observations in the pre-outbreak and current timeframes across vast swatches of territory (e.g. 9 observations in southeast Alaska pre-outbreak, 4 in western Gulf of Alaska currently). Additionally, in regions such as British Columbia observations are tightly clustered in just a few places and provide limited coverage of the entire region. Thus, most regions could not be meaningfully modeled across both time periods.

For these reasons, we cannot create meaningful regional models. However, one of the co-authors on this paper, Dr. Alyssa Gehman, is currently working on this exact analysis for parts of British Columbia using site-specific estimates of temperature at depth, wave exposure, water exchange, etc. We feel this kind of targeted, well-developed analysis is more appropriate for asking these questions at a regional scale. The results of the Gehman paper can then be used to confirm or complicate the findings of our paper.

In response to the reviewer's concern, however, we have added language in the Discussion that clarifies the limitations associated with our inferred relationship between temperature and SSWD (see responses to line comments 331-337). While we do not think it is a good idea to do the regional models suggested by the reviewer here, we hope our added discussion of the limitations of the model will ensure we do not overstate the results.

Comments:

Line 25. "We are left dramatically unprepared" is a departure from the focus of the rest of the paragraph. I suggest rephrasing.

Please see tracked changes for the rephrased sentence (line 25).

Line 52. In question #3 you ask how were the environmental variables associated with Pycnopodia distribution different before and after the outbreak instead of asking what conditions were associated with the timing of the declines that you are documenting, and I think many readers are going to wonder why you didn't ask the latter question. The points you make at the in the prior paragraph hint at this, but the introduction could use some language that pulls together why thus far it has been difficult to link environmental conditions directly to the timing of decline.

There are a few reasons that it will be difficult to explore the triggers of SSWD, particularly in Pycnopodia. Mortality in Pycnopodia happens incredibly rapidly following the first signs of disease, and thus even though we are working with the best data that exist on timing of disease, it is likely that we missed the exact timing of the outbreak in many locations. Thus, the timing of first emergence of the disease will be confounded with the timing of epidemic spread of the disease, making it impossible to disentangle the epidemic dynamics from environmentally triggered dynamics. For example, with SAR-COV-2, in the early phases of an epidemic the availability of susceptible hosts easily swamps environmental signals (e.g. see Baker et al. 2020). SDM's are a powerful tool when applied to disease ecology questions, but, as has been

demonstrated with the application to COVID-19, there must be careful alignment of the question with the disease in question (see Carlson et al. 2020: <https://www.nature.com/articles/s41559-020-1212-8>). By modeling the pre- and post-outbreak populations we feel we have best used the data in hand to evaluate the relationship between environmental drivers with changes in the species distribution rather than a relationship between environmental drivers and the disease itself. We have put in additional wording in the introduction to address the points made here (line 47 and 53).

Line 56-58. This is an impressive compilation of data.

Thank you!

Line 85. Why did you choose the 10th percentile as the cutoff for the phases? Please clarify the rationale.

Please see lines 95-98 for the added rationale.

Line 90. Similar to above, why did you choose a 75% decline as the threshold for the population crash? Is this rooted in the species' population dynamics or ability to perform its function as a predator?

Please see lines 102-104 for the added rationale.

Line 127. Briefly list which variables were highly correlated and removed. How did you choose which variable to eliminate? For example, 90th percentile temperature is likely highly correlated with latitude.

We had intended to include our justification for excluding latitude from the model in the Supplement, and thank the reviewer for noting our oversight. Briefly, we wanted to exclude collinear variables when possible (see advice of Merow et al (2013)). Because temperature and latitude were collinear we tested our model with both latitude and our temperature metric. We found that the Maxent models' fits dropped when substituting latitude for temperature or when adding latitude along with temperature to the model. Because it did not improve model fit over temperature and because we felt there was stronger prior evidence for the relationship between temperature and SSWD than latitude and SSWD, we dropped latitude and kept our temperature metric. We have now included this justification as well as a graph of the relationship between latitude and temperature in the Supplement.

Per the reviewer's note, we also added more to the Supplement regarding how we investigated collinearity during variable selection for the Maxent models (see lines 193 - 237).

Line 128. For environmental factors that change over time (e.g., temperature, chlorophyll), it would be useful to run the Maxent model using separate climatologies from the 2017-2020 period to evaluate the abiotic factors associated with the remaining population distribution. For example, if temperatures have increased substantially, using the 30-year mean temperature is no longer a reflection of the conditions Pycnophidia must deal with.

As noted above, this was an oversight on our part in the Methods section, and again thank the reviewer for their careful reading. To briefly summarize, we did calculate mean chlorophyll and 90th percentile of temperature for the 2009-2012 period for the pre-outbreak model and for the 2017-2020 period for the post-outbreak model, but failed to detail this in the paper. Salinity and substrate data, however, were not available at high enough temporal and spatial scales to create statistics for individual periods and thus we resorted to using climatologies. While the climatologies undoubtedly miss trends on smaller scales over time, we feel that overall they do adequately represent continental scale variability in these parameters.

In the manuscript itself we have now clarified that the temperature and chlorophyll metrics were created for both the 2009-2012 and 2017-2020 periods for the pre- and post-outbreak models

respectively (see lines 144-146). Further, we have added a section to the Supplement that explains why salinity and substrate data could not be modeled for individual time periods (lines 257-270).

Line 133. Running the Maxent model separately for each region in addition to the whole coast would help to account for the regional differences across this large geographic range. If the same abiotic variables emerged as important in most of the regions, it would increase confidence in what factors are truly associated with the shifts in distribution and what is merely a product of the major environmental and ecological variation between regions.

To briefly summarize our response to this laid out above, we feel that the spatial scale of the environmental data and the spotty coverage of the Pycnopodia observation data do not allow us to create meaningful regional-level models. Additionally, one co-author is currently working on a paper doing just this for British Columbia in a much more thorough, scale appropriate way than we could. This other paper will help confirm or complicate our findings at a continental scale.

However, we recognize the reviewer's point that this continental-scale approach misses crucial regional-level variability and trends. In response, we have added language to the Discussion that more fully discusses the limitation of this continental-scale analysis and call for further regional investigations in future research (lines 331-337).

Line 168. Why is your threshold for 'Rare' <30%, when your threshold for a crashing population was 25%? It would be more logical if thresholds of rarity and a crashing population were linked.

We thank the reviewer for this excellent point and have changed the analysis and associated figure to reflect this.

Line 204. "Still severe" would be a better descriptor than "somewhat less severe" in this case.

We agree and have changed the text to reflect this.

Line 232-233. Does using the 2017 - 2020 mean temperature instead of the 30-year mean influence this result, or does running a separate Maxent model for each region influence this result?

In regards to the Reviewer's question about using 2017-2020 temperature data, we believe we addressed this in our response to the line 128 comment. To reiterate, the temperature and chlorophyll metrics were derived using 2009-2012 data and 2017-2020 data for the pre-outbreak and current models respectively. Regarding the question about the regional models, please see our response to the Line 133 comment.

Line 275-287. Why does knowing the regional differences in timelines of the population crash matter? What do we gain from understanding it?

Thank you for highlighting this section, we agree that we had missed adding complete context in this section. We have added some additional information exploring the potential drivers of this variability, and highlight areas of future research (lines 301-304 and 312 – 314).

Line 289. Is there some way to separate out influence of temperature and latitude in the Maxent model? It is difficult to tell what role temperature versus latitude had in the shifts in occurrence probability. If a spatial pattern emerged where hotspots within a region had higher Pycnopodia decline than the area around it, that would be very compelling evidence. To borrow an example from a different sea star species, at this coast-wide scale, there was low evidence for temperature being associated with *Pisaster's* decline, even though lab experiments showed that high temperatures hasten the course of SSWD.

We addressed some of this point above in our response to Line 127 comments. Briefly, we did consider latitude when building these models but failed to add this information to the Methods. We eliminated latitude because 1) it was collinear with temperature, 2) substituting it into our models for temperature (or with temperature) decreased model fit, and 3) we felt that temperature had more biological relevance and previous evidence of a relationship to SSWD than latitude. We have now added this information to the Supplement. In addition, we have now added several sentences to the discussion that more clearly outline that latitude cannot be easily disentangled from temperature in this analysis and an alternative hypothesis for how latitude could be influencing the progression of SSWD.

Finally, we would like to point out that the papers on Pisaster that do not find a positive relationship with temperature (e.g. Miner et al. 2018, Menge et al 2016) all focus on the relationship between the initial emergence of SSWD symptoms and temperature, not the relationship between the population level severity of SSWD and temperature as we do here. In fact, in the excellent work by Miner et al (2018) Fig. 2 actually does show that southern populations were more negatively impacted by SSWD than northern populations. While again, this observation has the same problem that it conflates temperature with latitude, we don't believe our finding here actually conflicts with previous work done on Pisaster specifically because we were not associating temperature with SSWD symptom emergence.

Fig 3. Putting the data range under the panel title would improve this figure.

We agree and have revised the figure.

Supplemental Material

Line 230. There appears to be an empty table above table 4.

This caption refers to a very large table that takes up an entire page. We have rearranged the two so that the first half of the table immediately follows the caption for clarity.

Fig. S2. Are the arrows pointing to the epidemic phase timeline supposed to line up with the shifts in phase? It seems odd that they don't.

We apologize for this oversight and have adjusted this Figure.

Fig. S3. With the major differences in densities between the regions, this figure would be better split up into a series of panels so that the trends in each region can be examined. You can group them by larger geographic areas such as quarters of their range or individual regions.

We thank the reviewer for this suggestion and have made the advised change.

Referee: 2

Comments to the Author(s)

This manuscript compiles range-wide survey data on Pycnopodia to describe the recent progression of SSWD and population trends in the species. This compilation of data is striking and impactful all by itself. The authors also use species distribution models (SDMs) to generate hypotheses on potential environmental drivers of SSWD in this species. I will note that I am not expert in species distribution modeling, so I hope that another reviewer with that specific expertise evaluates that component of the manuscript.

I really have no major criticisms of the manuscript. The dataset is unique in scale, the analyses all seem appropriate, and the writing is clear and concise. I thank the authors for pulling all these data together; this is an important paper.

I do have a few minor suggestions for improvement, however, listed below.

1) The authors used data from both *Pisaster ochraceus* and *Pycnopodia* to identify the date SSWD was first observed (in *Pycnopodia*), noting in the methods that they did so because there are more observations for *Pisaster*, and because inspection of the data showed co-incident outbreaks between the two species. That second point is important, because the two species likely differ in distribution, with *Pisaster* being much more common in shallower water (and especially the intertidal zone), and *Pycnopodia* in the shallow and deeper subtidal. Given the potential importance of temperature in influencing SSWD, and systematic differences in temperature with depth, this might lead to some bias in the estimation of date SSWD was first observed, which would affect estimates of emerging epidemic and outbreak duration. I would suggest that the authors mention this possibility briefly in the manuscript, and provide at least some evidence in support of the “co-incident outbreaks between the two species” in the supplementary material.

We have now addressed this in the manuscript (lines 88-90) and have added evidence of co-incident outbreaks between the two species in the supplement (Fig. S3).

2) The authors treat the recent epidemic, which started in 2013, as unique (e.g., in the introduction, “The SSWD epidemic... began in 2013.”) But there have been earlier outbreaks of disease among west coast seastars, including *Pycnopodia*, that are indistinguishable in symptoms from those of the current epidemic. E.g., see Eckert et al. 1999, Sea star disease and population declines at the Channel Islands. Proc 5th CA Channel Islands Symp. 1999;5: 390–393 (pdf freely available online). We don’t know the etiological agent(s) of the current or prior epidemics, so cannot say that the causes are the same. However, it would be nice to note somewhere in the introduction or discussion that the current epidemic is not necessarily a unique event in the history of *Pycnopodia* or west coast seastars more generally (though it may be greater in intensity and spatial extent than prior epidemics).

This is a great point, and we have added a sentence discussing this in the Introduction on line 30-32.

One other related issue is that prior epidemics (e.g., 1982-1984, and 1997-1998) may have had some effect on historic population sizes. Such potential effects are not visible from the data presented, because though the authors collected survey data spanning 1967 to 2020, they only illustrate (in Fig. 1a) predicted incidence from 2012 to 2020. I don’t know why this is, though perhaps the older data are too patchy in space to generate predictions for all regions? If it is possible to generate predicted incidence all the way back to the late 1960s, that would be very interesting for readers to see, and would provide some very useful historical information on variation in population size and perhaps the timing of prior epidemics in this species.

We agree that this would be very interesting. Unfortunately, the reviewer is correct that the older data are too patchy to generate meaningful predictions. Only a single dataset goes back to 1967, and only 6 of the 30+ datasets goes back before 1990. We specifically limited the species distribution models to 2009-2012 and 2017-2020 because we wanted to minimize the impacts of differential sampling effort between the two periods. Maxent models are sensitive to uneven sampling effort, and we tried to minimize this whenever possible. We felt that including data from 1967 - 2012 for the pre-outbreak model would a) pit 35 years of sampling pre-SSWD against just 3 years of sampling post-outbreak and b) could bias models towards older data that was not

representative of Pycnopodia populations near the beginning of SSWD. We have added a note about this in the Supplement for readers in the future that may have this question.

3) A very minor point – Fig. 1b is cited in the results before Fig. 1a. Why not simply switch those two so you can cite figures in order? I suspect this was a minor error introduced in revision, because in the second paragraph of the discussion the authors refer to Fig 1b when I think they mean to refer to the current Fig. 1a (which really should be b). The current Fig. 1b does not illustrate the point they are trying to make in this paragraph.

We thank the reviewer for their careful review and have altered Fig. 1 as suggested.